# Guard cells control hypocotyl elongation through HXK1, HY5, and PIF4

Gilor Kelly [1], Danja Brandsma[1], Aiman Egbaria[2], Ofer Stein[1], Adi Doron-Faigenboim[1], Nitsan Lugassi[1], Eduard Belausov[1], Hanita Zemach[1], Felix Shaya[1], Nir Carmi[1], Nir Sade[2] & David Granot[1✉]

The hypocotyls of germinating seedlings elongate in a search for light to enable autotrophic sugar production. Upon exposure to light, photoreceptors that are activated by blue and red light halt elongation by preventing the degradation of the hypocotyl-elongation inhibitor HY5 and by inhibiting the activity of the elongation-promoting transcription factors PIFs. The question of how sugar affects hypocotyl elongation and which cell types stimulate and stop that elongation remains unresolved. We found that overexpression of a sugar sensor, Arabidopsis hexokinase 1 (*HXK1*), in guard cells promotes hypocotyl elongation under white and blue light through PIF4. Furthermore, expression of *PIF4* in guard cells is sufficient to promote hypocotyl elongation in the light, while expression of *HY5* in guard cells is sufficient to inhibit the elongation of the *hy5* mutant and the elongation stimulated by *HXK1*. HY5 exits the guard cells and inhibits hypocotyl elongation, but is degraded in the dark. We also show that the inhibition of hypocotyl elongation by guard cells' HY5 involves auto-activation of *HY5* expression in other tissues. It appears that guard cells are capable of coordinating hypocotyl elongation and that sugar and *HXK1* have the opposite effect of light on hypocotyl elongation, converging at PIF4.

---

[1] Institute of Plant Sciences, Agricultural Research Organization, The Volcani Center, Rishon LeZion, Israel. [2] School of Plant Science and Food Security, Tel Aviv University, Tel Aviv, Israel. ✉email: granot@agri.gov.il

The growth of seedlings from germinating seeds is dependent on the conversion of seed carbon reserves to sugar, usually sucrose, and the movement of that sugar to the developing hypocotyl and cotyledons[1–3]. The hypocotyls of germinating seeds elongate in the dark, in a search for light that will allow the cotyledons to switch from relying on reserve generated sugars to autotrophic photosynthetic sugar production. Stomata, composed of two guard cells, appear on the cotyledons soon after germination, to allow gas exchange for photosynthesis and sugar production, once the seedlings reach the light[4,5]. In dark-grown Arabidopsis seedlings, CONSTITUTIVELY PHOTO-MORPHOGENIC 1 (COP1, an E3 ubiquitin ligase)/SUPPRESSOR OF PHYA (SPA) and DE-ETIOLATED 1 (DET1) complexes act as master suppressors of light signaling[6–8]. These complexes target elongation suppressors such as ELONGATED HYPOCOTYL5 (HY5) for degradation, thereby enabling hypocotyl elongation[9–11]. The hypocotyl elongation is mediated by PHYTOCHROME INTERACTING TRANSCRIPTION FACTORs (PIFs), which stimulate the production of the auxin that is necessary to promote hypocotyl elongation[11,12]. Light halts hypocotyl elongation through the activation of blue-light photoreceptors called cryptochromes (CRY) and red-light photoreceptors called phytochromes (PHY)[13]. Light-activated photoreceptors interfere with the activity of the COP1/SPA and DET1 complexes, preventing the degradation of the hypocotyl elongation inhibitor HY5 and blocking the transcriptional activity of PIFs, thereby inhibiting hypocotyl elongation[6,10,12,14–19].

Shade (a low ratio of red/far red light) and low levels of blue light also promote hypocotyl elongation via PIFs[20,21]. Those conditions are sensed by the photoreceptors in the cotyledons, which drive the synthesis of auxin, which is transported to the hypocotyl to induce hypocotyl elongation[22–26]. In the hypocotyl, auxin stimulates the synthesis of brassinosteroids (BR) and the responses that are required for hypocotyl elongation[27,28]. Evidence suggests that the epidermis is a central player in the regulation of hypocotyl elongation. The hypocotyls of photochrome B mutants (phyB) elongate in red light and expression of PHYB in the epidermis of phyB mutant is sufficient to prevent elongation in red light, suggesting that red-light perception occurs in the epidermis and prevents both auxin production and the transport of auxin out of the cotyledon[29]. Suppression of BR biosynthesis in the epidermis restricted shoot growth; whereas restoring BR biosynthesis or BR receptors in the epidermis of BR mutants rescued the dwarf phenotype[28]. High temperatures also cause hypocotyl elongation and the epidermis coordinates thermoresponsive growth through the PHYB-PIF4-auxin pathway[30]. These studies suggest that the epidermis plays a role in hypocotyl elongation. However, the epidermis is composed of epidermal pavement and guard cells and it is not clear which of those two types of cells is involved in the elongation signals.

In addition to light, the role of sugars in hypocotyl elongation has also been studied. Sucrose and glucose have been shown to stimulate auxin biosynthesis in Arabidopsis and several studies have observed that external sugars such as sucrose or glucose promote hypocotyl elongation, which is mediated by PIF transcription factors and auxin[31–37]. Sucrose is a disaccharide that must be cleaved, yielding the hexose monomers glucose and fructose, which must be phosphorylated before they can be metabolized[38]. There are only two distinct groups of enzymes that can phosphorylate the glucose and fructose: hexokinases (HXK) and fructokinases (FRK)[38]. HXKs phosphorylate both glucose and fructose; whereas FRKs are specific to fructose[39]. HXK is an important sugar sensor that monitors sugar levels in various tissues, in addition to its enzymatic activity[40–43]. For example, the Arabidopsis HXK1 inhibits the expression of photosynthetic genes in mesophyll cells in response to high sugar levels[40–42], and HXK1 within guard cells controls guard-cell behavior and reduces stomatal apertures in response to high sugar levels, thereby coordinating sugar production with transpiration[43–46]. Previous studies have examined the behavior of the HXK1 mutant (gin2) in terms of hypocotyl elongation and auxin levels in response to sugar treatments and reached ambiguous conclusions, perhaps due to redundant activity with the other HXKs, HXK2, and HXK3[34–38].

In this study, we took a different approach in which we overexpressed HXK1. We found that expression of HXK1, either globally or only in guard cells, stimulates hypocotyl elongation under long-day conditions. This HXK1-induced elongation is mediated by COP1, PIF4 and auxin signals, and competes with the effects of light and HY5. Furthermore, we show that increased expression of PIF4 or HY5 only in guard cells is sufficient to promote or inhibit hypocotyl elongation, respectively. Thus, HXK1 mediates the effects of sugar on hypocotyl elongation and guard cells alone are capable of controlling hypocotyl elongation.

## Results

**Hexokinase promotes hypocotyl elongation.** Sucrose promotes hypocotyl elongation in the presence of light[32,33,35–37]. Indeed, the hypocotyls of Arabidopsis seedlings grown under long-day conditions (16 h light/8 h dark, 40 µE) on plates containing 1% sucrose elongated about 20% further than the hypocotyls of similar seedlings grown on control plates without sucrose (Fig. 1b–d). To explore the role of the known sugar sensor HXK1

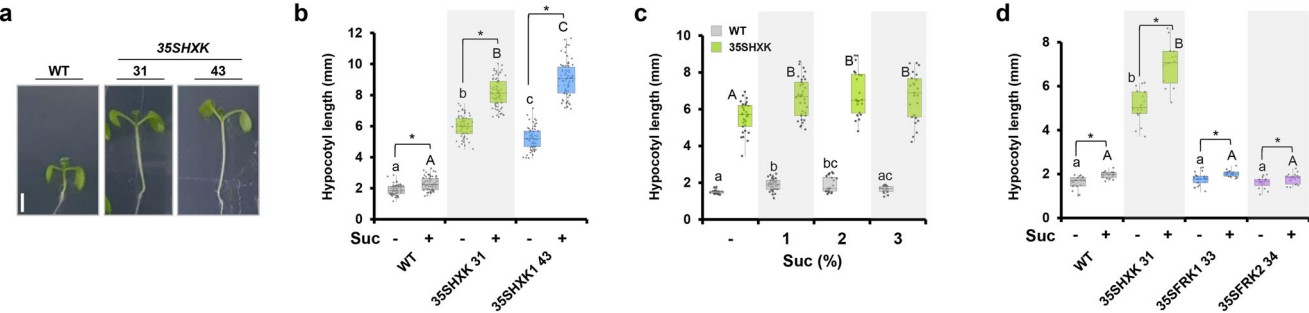

**Fig. 1 Hexokinase promotes hypocotyl elongation. a–d** Seedlings were grown under long-day (16 h light/8 h dark), white-light (40 µmolm$^{-2}$ s$^{-1}$) conditions for 7 days. **a** Representative images of 7-day-old 35SHXK seedlings. Bar = 2 mm. **b** Hypocotyl lengths of 35SHXK seedlings grown with and without 1% sucrose. Data from two independent lines (#31, #43) are shown. **c** Hypocotyl lengths of 35SHXK seedlings treated with increasing concentrations of sucrose. **d** Hypocotyl lengths of 35SHXK, 35SFRK1 and 35SFRK2 seedlings grown with and without 1% sucrose. **b–d** The box plots extend from the first to third quartiles and the whiskers extend from the minimum to the maximum levels. Lines within the boxes signify median values and light gray dots represent individual data points ($n = 55$–70 for **b**; $n = 20$–30 for **c**; $n = 15$–25 for **d**). Different lower-case and capital letters indicate significant differences (Tukey's HSD test, $P < 0.05$). Asterisks indicate significant differences between the compared treatments ($t$ test, $P < 0.05$).

in sucrose-induced hypocotyl elongation, we measured the length of the hypocotyls of 7-day-old seedlings of independent lines overexpressing *HXK1* under the 35S global promoter (35SHXK plants). Under long-day conditions, two previously described independent lines, 35SHXK31 and 35SHXK43, with high levels of *HXK1* expression[47], had hypocotyls that were about 3 times longer than those of WT seedlings (Fig. 1). Supplementation of the medium with 1% sucrose promoted the elongation of the 35SHXK lines about 35% further, suggesting that the sucrose-induced elongation effect is mediated by HXK1 (Fig. 1b–d). Higher concentrations (2–3%) of sucrose affected elongation much like 1% sucrose (Fig. 1c). Hypocotyl elongation was not observed in Arabidopsis plants overexpressing fructokinases, either *FRK1* or *FRK2*, which are fructose-specific phosphorylating enzymes distinct from HXK (Fig. 1d)[38,48]. This suggests that the elongation observed in 35SHXK is specific to HXK and is not a metabolic or a general hexose-phosphorylation effect.

**Hexokinase in guard cells is sufficient to promote hypocotyl growth**. While a few studies have suggested that the epidermis affects hypocotyl elongation by generating hormonal cues[22,23,27,28,30], the contribution of guard cells per se to hypocotyl elongation is not known. Stomata, each composed of two guard cells, appear on the epidermis of the cotyledons immediately after the emergence of the cotyledons from the seed coat[5]. To examine the involvement of guard cells in hypocotyl elongation, we used seedlings that express *HXK1* specifically in their guard cells (named GCHXK plants for guard-cell HXK)[44]. Similar to 35SHXK seedlings, the hypocotyls of independent GCHXK lines (GCHXK2 and GCHXK4) were about three times longer than those of WT seedlings under long-day conditions (Fig. 2a–c), indicating that guard cells are sufficient to trigger HXK-mediated hypocotyl elongation. The hypocotyls of GCHXK2 seedlings were longer than those of GCHXK4 seedlings and the hypocotyls of the offspring of the cross between the two lines (referred to as GCHXK24) were slightly longer than those of the GCHXK2 seedlings, indicating a near maximal elongation effect of GCHXK (Fig. 2a). Like 35SHXK, the presence of 1% sucrose caused the hypocotyls to elongate significantly further (Fig. 2a). A day-by-day analysis showed that elongation of WT seedlings ceased almost entirely 2 days after germination, while that of GCHXK seedlings continued up to 6 days after germination (Fig. 2b). These results suggest that HXK stimulates elongation by extending the hypocotyl growth period, in line with an earlier study showing that sucrose extends the growth period of WT seedlings[32]. Scanning electron microscopy (SEM) has indicated that the hypocotyl elongation of GCHXK is due to cell elongation, as is the case in regular hypocotyl elongation (Fig. 2d, e)[32].

To explore the molecular response triggered by GCHXK, we performed a transcriptomic analysis of seedlings grown under long-day conditions (16 h light/8 h dark, 40 µE), 4 days after germination (Supplementary Data 1, 2). We identified 1011 differentially expressed genes (DEGs) that were downregulated (843 genes) or upregulated (168 genes) by GCHXK (Supplementary Figs. 1, 2, 3, 4). The pathways containing the highest number of DEGs are primarily associated with metabolic activity, the biosynthesis of secondary metabolites and carbon metabolism (Supplementary Fig. 2). Among the metabolic processes affected by GCHXK, a downregulation of photosynthesis-related pathways stood out (Supplementary Fig. 1). The transcript levels of genes involved in the light reaction step, as well as those of genes associated with the Calvin–Benson cycle pathway and carbonic anhydrases, were significantly lower, as also seen in the functional overview (Supplementary Fig. 1, 1-photosynthesis). Inhibition of photosynthesis during hypocotyl elongation was reported previously[9,17] and it appears that the hypocotyl elongation promoted by GCHXK has a similar effect.

A qPCR analysis of hypocotyl elongation-related genes verified that the expression of *HY5*, a suppressor of hypocotyl elongation in the light, was lower in GCHXK plants. The expression of *COP1* and *PHYB* remained unchanged, but the expression of *DET1* (a repressor of HY5) and of *PIF1,3,4*, which promote hypocotyl elongation in the dark[12], were higher (Fig. 3a, c). Along with the upregulation of PIFs, the expression of the auxin-responsive genes *SMALL AUXIN UP RNA 50, 65* (*SAUR50, SAUR65*) and genes involved in cell elongation [*XYLOGLUCAN ENDOTRANSGLUCOSYLASE-RELATED PROTEIN7* (*XTR7*) and EXPANSIN8 (*EXP8*)], whose expression is stimulated by PIF[49,50], was also upregulated (Fig. 3b). The auxin biosynthesis gene *YUCCA8* (*YUC8*) that promotes elongation in the dark[49] was upregulated as well (Fig. 3b). Accordingly, auxin levels in GCHXK seedlings were higher than those in WT seedlings (Fig. 3d) and inhibiting auxin transport via the polar auxin transport inhibitor N-1-naphthylphthalamic acid (NPA) prevented the elongation of the hypocotyls of GCHXK seedlings (Fig. 3e). These results support the notion that GCHXK promotes hypocotyl elongation under light conditions via the known elongation-related molecular pathways that stimulate auxin production and signals.

**GCHXK stimulates elongation under blue light**. Since either blue or red light can inhibit hypocotyl elongation, we tested whether GCHXK elongation under white light overcomes the inhibition caused by blue or red light. The hypocotyls of GCHXK seedlings grown in blue light (16 h light/8 h dark, 14 µE) were 4 times longer than those of WT seedlings, indicating that sucrose and GCHXK can overcome the blue-light inhibition of hypocotyl elongation (Fig. 2f). However, GCHXK failed to stimulate hypocotyl elongation when seedlings were grown in red light (16 h light/8 h dark). Under a low red-light intensity of 40 µE, the hypocotyls of both WT and GCHXK were similarly elongated; whereas at 130 µE, the hypocotyls of both lines were relatively short (Fig. 2g). We concluded that GCHXK overcomes the elongation–inhibition effect of blue light, but not that of red light. Below, we discuss the biological meaning of the different effects of blue and the red light on GCHXK elongation.

**Sucrose and hexokinase in guard cells act upstream of PIF4 and compete with HY5**. To explore the interplay between sucrose, HXK and various genes involved in hypocotyl elongation, we analyzed the effect of sucrose on elongation-related mutants. The hypocotyls of *hy5* mutant elongate in the presence of light and elongate even further in the presence of sucrose (Fig. 4a). These results suggest that the additive elongation promoted by sucrose does not require the suppression of *HY5* by sucrose, as this gene was already mutated. Rather, it suggests that the promotion of hypocotyl elongation by sucrose occurs via an independent pathway, which probably competes with the inhibitory effects of HY5. Unlike *hy5* mutant, *cop1-4* mutants, which do not elongate under dark or light conditions, do not elongate any further in presence of sucrose; whereas the *pif4* mutant and the triple *pif3-5* mutant exhibit slightly more elongation in the presence of sucrose (Fig. 4a). These results support the notion that sucrose enhances elongation via COP1. The fact that sucrose affects elongation independent of HY5, but fails to promote hypocotyl elongation in the *cop1* mutant indicates that COP1 activates hypocotyl elongation not only by targeting HY5 for degradation, but also via additional effects on other transcription factors, a notion that has already been proposed previously[6].

To further explore the interplay of GCHXK with HY5 and PIF4, we studied *hy5* and *pif4* mutants expressing GCHXK: the GCHXK/*hy5* and GCHXK/*pif4* lines, which were obtained by

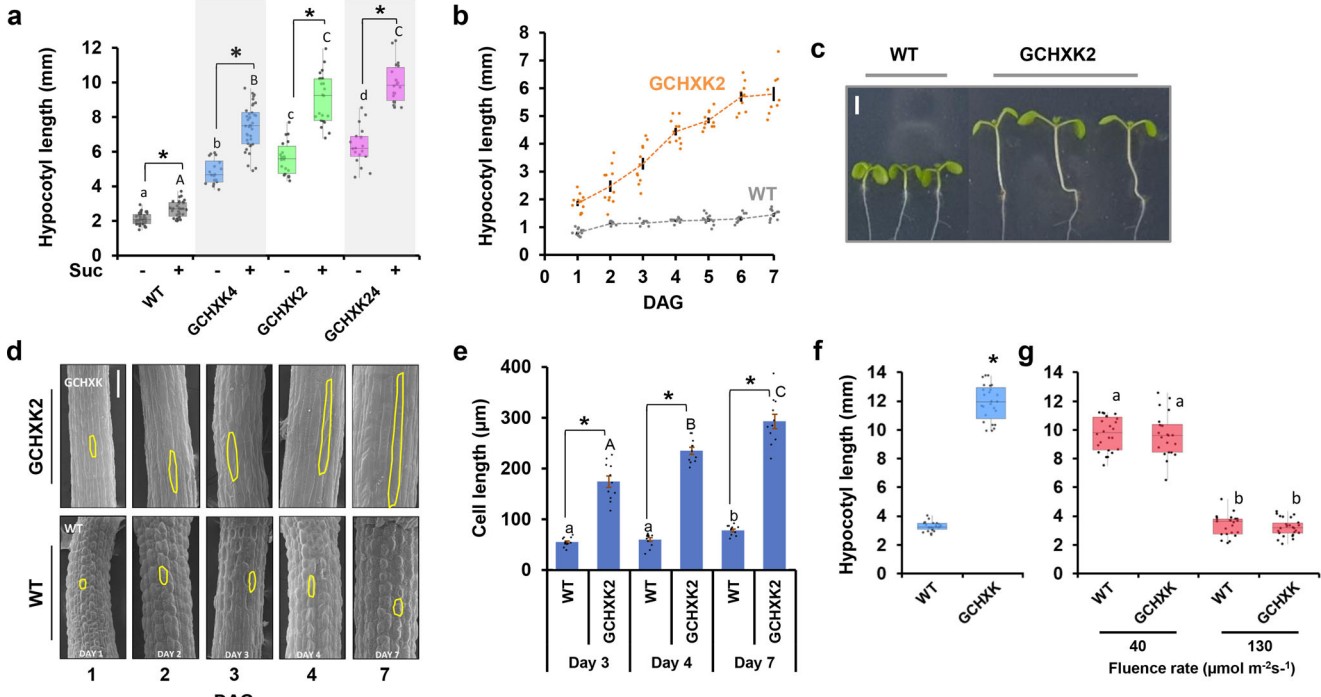

**Fig. 2 Guard-cell expression of hexokinase is sufficient to promote hypocotyl elongation. a–g** Seedlings were grown under long-day (16 h light/8 h dark, 40 μE) conditions for 7 days on 0.5× MS agar medium. **a** Hypocotyl length of *GCHXK* seedlings grown with and without 1% sucrose. Two independent lines (GCHXK2, GCHXK4) and homozygote offspring of the cross between those two lines (GCHXK24) are shown. **b** Hypocotyl lengths of GCHXK2 and WT seedlings measured for 7 days after germination. Data are means ± SE (n ≥ 7). Orange and gray dots represent individual data points for GCHXK2 and WT, respectively. **c** Representative images of 7-day-old WT and GCHXK2 seedlings; bar = 2 mm. **d** SEM images of WT and GCHXK hypocotyls taken at Days 1–4 and 7 following germination. Representative cell borders are highlighted, Bar = 100 μm. **e** Average cell length of WT and GCHXK epidermal cells at days 3, 4, and 7 following germination. Data are means ± SE (n ≥ 12). Gray dots represent individual data points. **f, g** Hypocotyl lengths of WT and GCHXK seedlings under (**f**) blue and (**g**) red light conditions with 1% sucrose. Light intensities were kept at 14 μE for blue light and 40 or 130 μE for red light. **a, e, g** Different lower-case and capital letters indicate significant differences (Tukey's HSD test, P < 0.05). Asterisks indicate a significant difference relative to the control (**a**) or relative to WT (**e, f**, t test, P < 0.05). **a, f, g** The box plots extend from the first to third quartiles and the whiskers extend from the minimum to the maximum levels. Lines within the boxes signify median values and light gray dots represent individual data points (n = 20–35).

crossing the GCHXK2 line with *hy5* and *pif4*, respectively. In the light, GCHXK/*hy5* elongated further than GCHXK or *hy5* single mutants did (Fig. 4b). The additive elongation effect of GCHXK over that of the *hy5* mutant further suggests that GCHXK stimulates hypocotyl elongation via a pathway that is independent of HY5 (i.e., it is not exerted through the suppression of HY5, but rather competes with the inhibitory effects of HY5). Unlike GCHXK/*hy5*, the hypocotyls of GCHXK/*pif4* failed to elongate, indicating that PIF4 is central for GCHXK elongation and that GCHXK acts upstream of PIF4 (Fig. 4c).

The DET1 complex is needed to target HY5 for degradation and the hypocotyls of the *det1* mutant did not elongate, even under dark conditions[6]. We wondered whether the elongation observed in GCHXK (under light) operates through DET1. To explore that issue, we constructed plants that expressed GCHXK in the background of *det1* (GCHXK/*det1*) by crossing GCHXK2 with the *det1* mutant. While the hypocotyl elongation of GCHXK seedlings increased about threefold, it was abolished entirely in GCHXK/*det1*, where it was similar to that observed for *det1* alone (Fig. 4d). These results suggest that DET1 is essential for the hypocotyl elongation of GCHXK seedlings. The dominant inhibitory effects of *det1* and *pif4* on GCHXK hypocotyl elongation demonstrate that the hypocotyl elongation observed among the GCHXK seedlings involved primarily DET1 and PIF4. The fact that GCHXK had an elongation effect even in the presence of HY5 (Fig. 4b), but failed to promote elongation of the *det1* mutant indicates that DET1 activates elongation not only by assisting in targeting HY5 for degradation,

but perhaps also by other previously proposed roles of DET1, such as chromatin regulation that affects gene expression[6].

**Expression of PIF4 and HY5 in guard cells affects hypocotyl elongation.** The results so far suggest that *HXK1* promotes hypocotyl elongation and that expression of *HXK1* specifically in guard cells is sufficient to stimulate that elongation. These results raise the possibility that signals generated exclusively within guard cells can control hypocotyl elongation. To examine this possibility, we expressed the transcription factors *PIF4* and *HY5* specifically in guard cells (GCPIF4 and GCHY5 lines, respectively) and examined their effects on hypocotyl elongation (Fig. 5). Exclusive expression of *PIF4* in guard cells of WT plants (GCPIF4 plants) promoted hypocotyl elongation in the light (increase of 30 to 67%) relative to the WT (Fig. 5a). Similar behavior was observed when GCPIF4 was expressed in the background of the *pif4* mutant (GCPIF4/*pif4*), in which the short hypocotyls of the *pif4* mutants (1.4 mm) were elongated by GCPIF4 (GCPIF4/*pif4*) to about 2 mm (Fig. 5b). Elongation was observed in all of the 20 independent lines tested, with the increases in length ranging from 18 to 55% (Fig. 5b, d). We concluded that the expression of *PIF4* in guard cells promotes hypocotyl elongation under light conditions.

With regard to HY5, hypocotyls of *hy5* mutants were 3 times longer than WT hypocotyls (6.7 mm vs. 1.76 mm) and global overexpression of *HY5* under the 35S promoter (35SHY5) completely abolished the long hypocotyls of *hy5* (Fig. 5c).

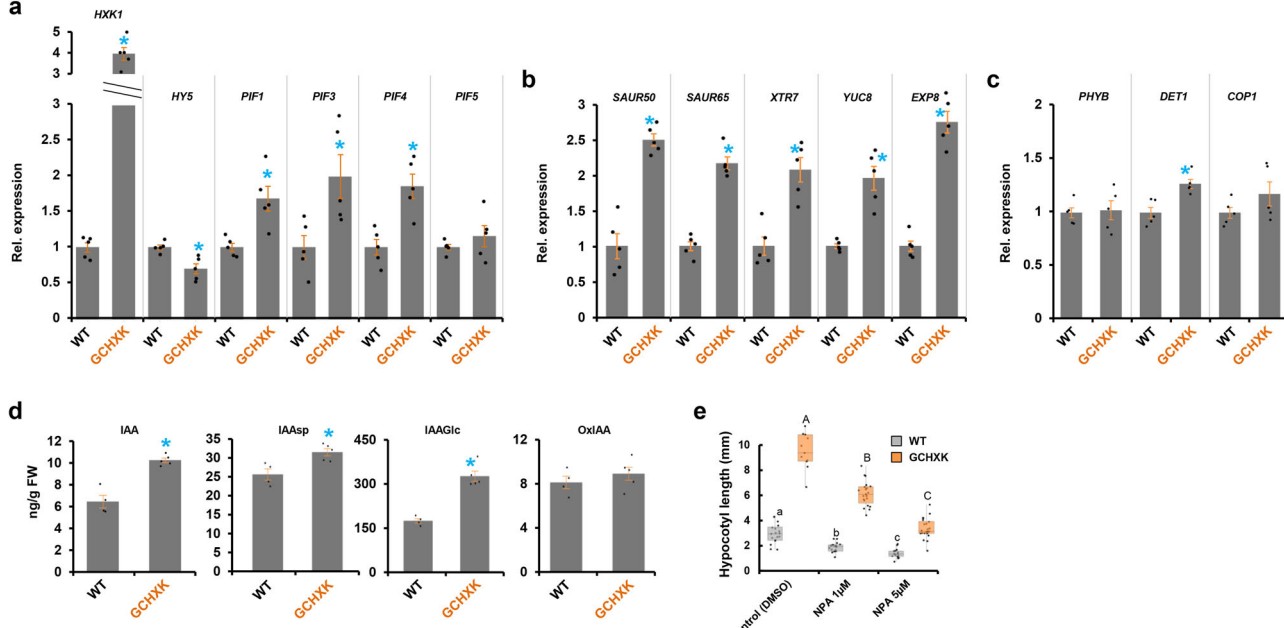

**Fig. 3 Relative expression of genes related to hypocotyl elongation, auxin levels, and auxin transport activity in GCHXK seedlings. a–c** RT-PCR expression analysis of genes related to the hypocotyl elongation of WT and GCHXK seedlings grown in 0.5 MS agar media. *TUB2* (β-tubulin) was used for normalization and the expression level in the WT was set to one. Data points are mean ± SE (*n* = 5). **d** GCHXK promotes the accumulation of auxin. IAA, IAAsp, IAGlu, and OxIAA quantification in 5-day-old WT and GCHXK seedlings. Data points for are means ± SE (*n* = 4 and 5 for WT and GCHXK, respectively). **a–d** Black dots represent individual data points, and blue asterisks indicate a significant difference relative to the WT (*t* test, *P* < 0.05). **e** Auxin transport activity is required for hypocotyl growth of GCHXK. WT and GCHXK seedlings grown in 0.5× MS agar media containing 1 or 5 μM of NPA for 9 days. The box plots extend from the first to third quartiles and the whiskers extend from the minimum to the maximum levels. Lines within the boxes signify median values and light gray dots represent individual data points (*n* = 12–25). Different lower-case and capital letters indicate significant differences (Tukey's HSD test, *P* < 0.05).

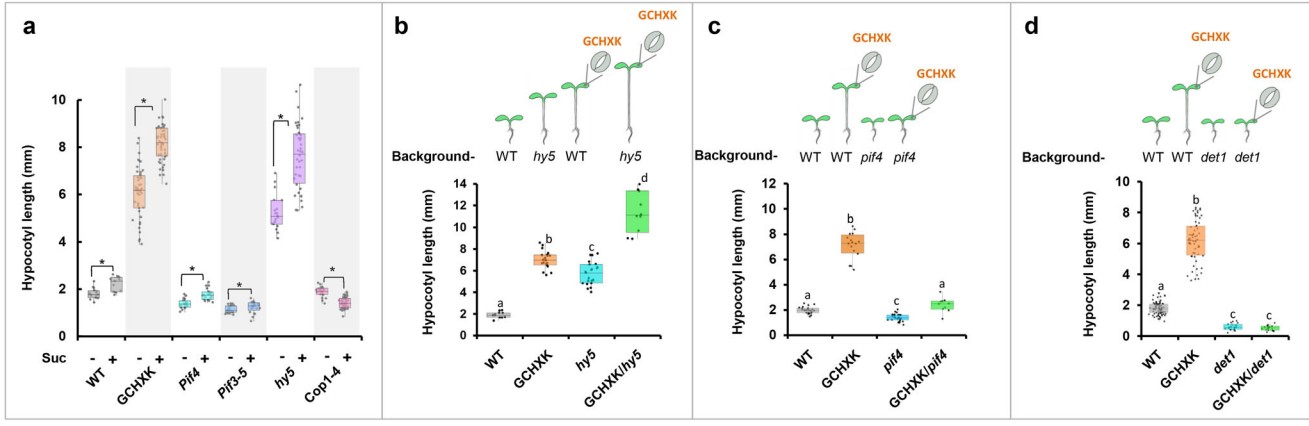

**Fig. 4 Hypocotyl lengths of mutants treated with sucrose and the combined effects of GCHXK with *hy5*, *pif4*, and *det1* mutants. a** Hypocotyl lengths of 7-day-old WT, GCHXK, *pif4*, *pif3-5*, *hy5*, and *cop1-4* seedlings grown under long-day conditions with or without 1% sucrose. **b–d** Hypocotyl lengths of (**b**) GCHXK/*hy5*, (**c**) GCHXK/*pif4*, and (**d**) GCHXK/*det1* seedlings grown on 0.5× MS agar media with 1% sucrose. **a–d** The box plots extend from the first to third quartiles and the whiskers extend from the minimum to the maximum levels. Lines within the boxes signify median values and gray dots represent individual data points (*n* ≥ 10 for **a–c**; *n* = 20–40 for **d**). **a** Asterisks indicate a significant difference relative to the control (*t* test, *P* < 0.05). **b–d** Different letters indicate a significant difference (Tukey's HSD test, *P* < 0.05). The illustration above each figure indicates the gene expressed in guard cells, the genomic background (WT, *hy5*, *pif4*, or *det1*) and the hypocotyl-growth response.

Similarly, exclusive expression of *HY5* in the guard cells of the *hy5* mutant (GCHY5/*hy5* plants) partially to fully inhibited the *hy5* elongated phenotype, reducing the hypocotyl lengths of independent GCHY5/*hy5* lines (Fig. 5c, d). These results from GCPIF4 and GCHY5 plants demonstrate that guard cells alone are capable of influencing hypocotyl elongation and that signals generated within the guard cells are exported and affect other tissues involved in the elongation process.

**GCHY5 counteracts the effect of GCHXK in an AtHY5-dependent manner.** As shown above, expression of *HXK1* in guard cells (GCHXK plants) in a WT background is dominant over the inhibitory effect of the endogenous HY5 and stimulates elongation in the light (Fig. 2); whereas the expression of *HY5* in guard cells in the background of the *hy5* mutant partially inhibits elongation (Fig. 5c). To better understand the interplay between HXK1 and HY5 in guard cells, we generated plants that co-

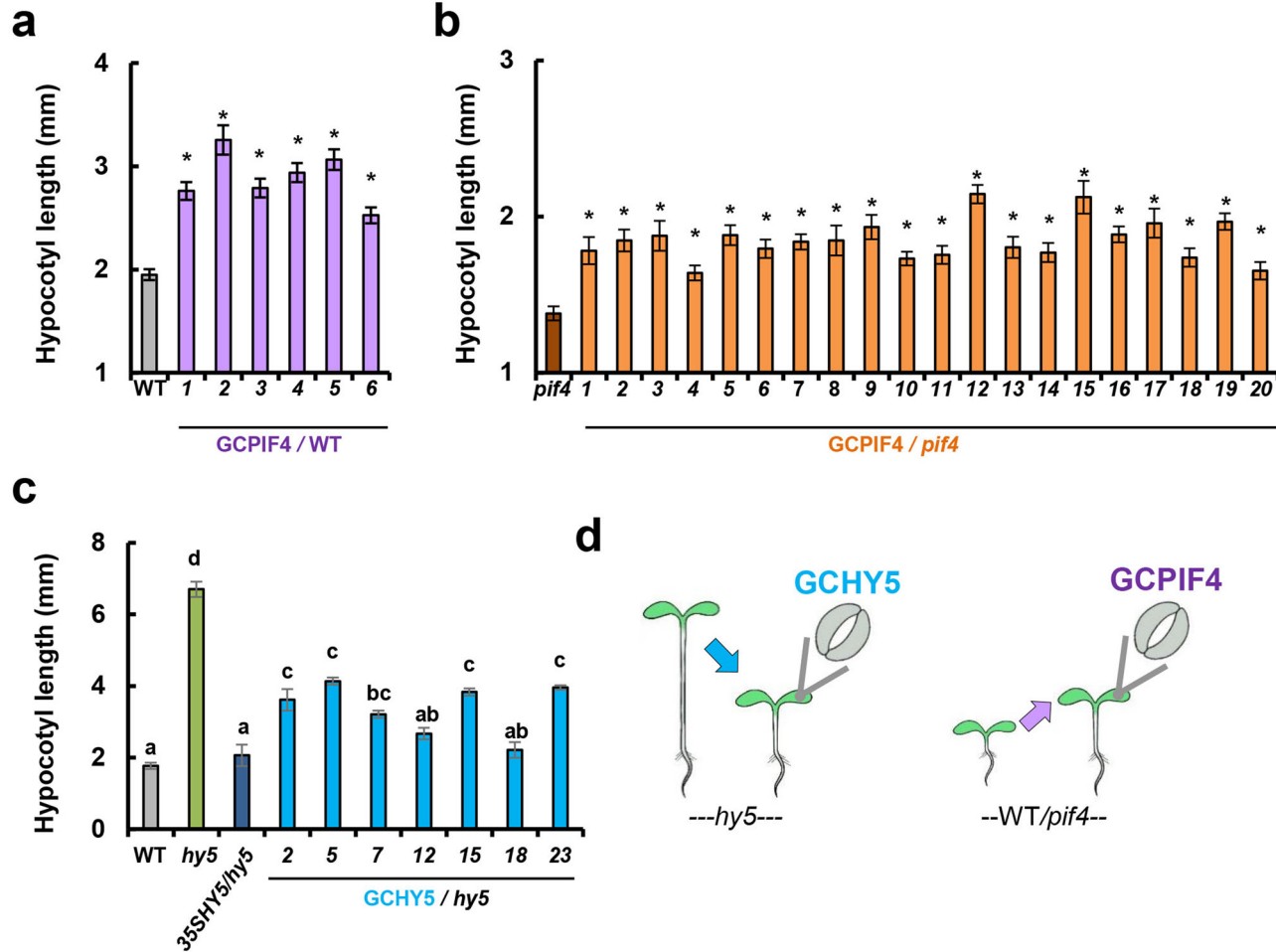

**Fig. 5 Expression of *PIF4* (GCPIF4) and *HY5* (GCHY5) in guard cells affects hypocotyl elongation. a** GCPIF4 in the WT background (purple columns) stimulates hypocotyl elongation of independent GCPIF4/WT lines relative to the WT. **b** GCPIF4 in the *pif4*-mutant background (orange columns) stimulates hypocotyl elongation of independent GCPIF4/*pif4* lines. **c** Hypocotyl lengths of independent lines expressing *HY5* in the guard cells (GCHY5/ *hy5*) or globally under the 35S promoter (35SHY5/*hy5*) in the *hy5*-mutant background (light-blue columns). Data points for (**a**–**c**) are means ± SE ($n = 20$–25 for **a**; $n = 15$–25 for **b**; $n = 13$–25 for **c**). **a, b** Asterisks indicate significant differences relative to the WT (Dunnett's test, $P < 0.05$). **c** Different letters indicate a significant differences (Tukey's HSD test, $P < 0.05$). **d** The illustration indicates the gene expressed in guard cells (GCHY5 or GCPIF4) and the genomic background (*hy5*, WT/*pif4*). The blue and purple arrows indicate the hypocotyl-growth response stimulated by GCHY5 and GCPIF4, respectively.

expressed *HXK1* and *HY5* in their guard cells (GCHXK/GCHY5 plants). These plants were created by crossing GCHXK2 with GCHY5/WT (WT background) and with GCHY5/*hy5* (*hy5*-mutant background). The hypocotyls of GCHXK/GCHY5 with the WT background (GCHXK/GCHY5/WT) were shortened from about 7 mm in GCHXK to only 2 mm, indicating that expression of *HY5* in guard cells is dominant over the effect of GCHXK (Fig. 6a). However, the hypocotyls of GCHXK/GCHY5 with the *hy5*-mutant background (GCHXK/GCHY5/*hy5*) were long (Fig. 6a), indicating that having HY5 only in the guard cells is insufficient to inhibit the elongation induced by GCHXK, and that the inhibition of elongation by GCHY5 probably requires the presence of HY5 in tissues other than guard cells.

HY5 is a shoot-to-root mobile transcription factor and HY5-GFP expressed under mesophyll- and phloem-specific promoters accumulates in the roots[51]. It has also been shown that HY5 protein binds to the *HY5* promoter and auto-activates its expression[51,52]. We, therefore, assumed that HY5 exits the guard cells of seedlings expressing GCHY5 and auto-activates the expression of HY5 in other tissues. However, guard cells do not have plasmodesmata and are symplastically isolated[53–55]. We,

therefore, decided to take a closer look at the question of whether HY5 is capable of exiting the guard cells. We generated transgenic plants expressing GFP fused to the HY5 protein under the control of the guard cell-specific promoter (GCHY5-GFP plants) and searched for GFP signal in tissues other than guard cells (Fig. 7). In control plants in which GFP was expressed in guard cells (GCGFP), the GFP-fluorescence signal was restricted to the guard cells and was not detected anywhere else (Fig. 7a)[5]. However, in the GCHY5-GFP plants (in which HY5 was fused to GFP), GFP signal was also detected in the nuclei of mesophyll cells and in phloem cells in the root, indicating that HY5-GFP produced within guard cells is transported outside those cells and enters the nuclei of cells in other tissues (Fig. 7a, b). The presence of HY5-GFP outside the guard cells was observed in seedlings grown under light (long-day) conditions and in seedlings grown in the dark (Fig. 8b). Yet, the HY5-GFP signal in the seedlings grown in the dark was half as strong as that of the seedlings grown in the light (Fig. 8a–d). This result suggests that HY5-GFP exits the guard cells even in the dark, but is probably targeted for degradation by the proteasome. To examine whether HY5-GFP is degraded by the proteasome, we applied the proteosome inhibitor

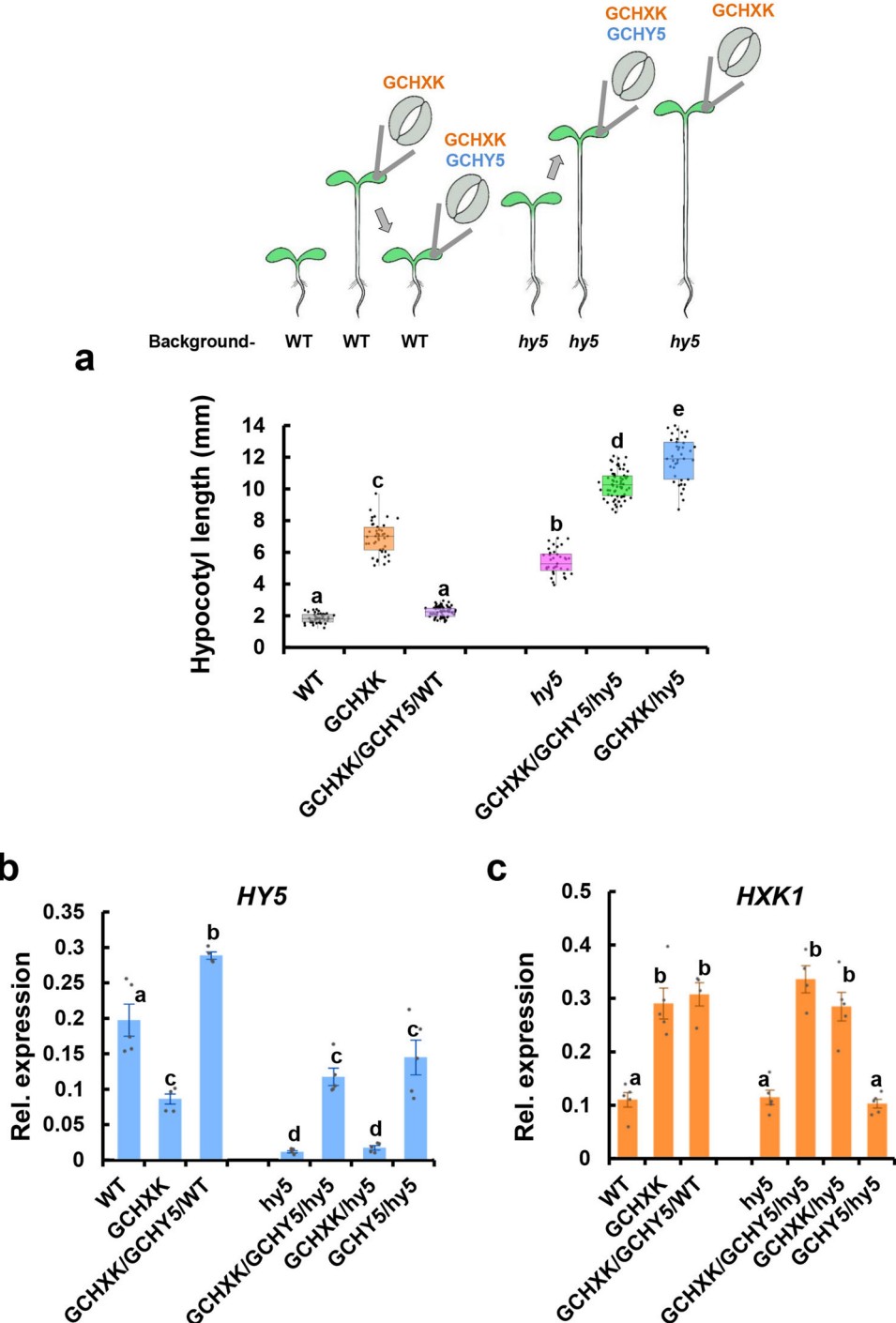

**Fig. 6 The ability of GCHY5 to reduce the hypocotyl elongation of GCHXK is dependent on the endogenous HY5. a** Hypocotyl lengths of GCHXK/
GCHY5/WT (WT background) and GCHXK/GCHY5/hy5 (hy5 background) seedlings grown with 1% sucrose. The box plots extend from the first to third
quartiles and the whiskers extend from the minimum to the maximum levels. Lines within the boxes signify median values and gray dots represent
individual data points (n = 30–60). The illustration at the top of the figure indicates the genes expressed in guard cells (GCHXK, GCHY5), the genomic
background (WT or hy5). Arrows indicate the hypocotyl-elongation response. **b, c** RT-PCR expression analysis of HY5 (**b**) and HXK1 (**c**) in GCHXK/
GCHY5/WT and GCHXK/GCHY5/hy5 seedlings. TUB2 (β-tubulin) was used for normalization. Data points are means ± SE (n = 5). Gray dots represent
individual data points. **a–c** Different letters indicate a significant difference (Tukey's HSD test, P < 0.05).

MG-132 to dark-grown seedlings and found that the level of
HY5-GFP signal was indeed higher in the presence of MG-132
(Fig. 8c, d). We concluded that HY5 exists in the guard cells in
the dark, but is then degraded and, therefore, fails to prevent
elongation. These results suggest that HY5 produced in guard

cells moves out of the guard cells and inhibits hypocotyl
elongation under light conditions by auto-activating the endo-
genous expression of HY5 in other tissues[51,52]. To examine this
assumption, we analyzed the expression levels of the endogenous
HY5 in 7-day-old seedlings. The expression level of HY5 in

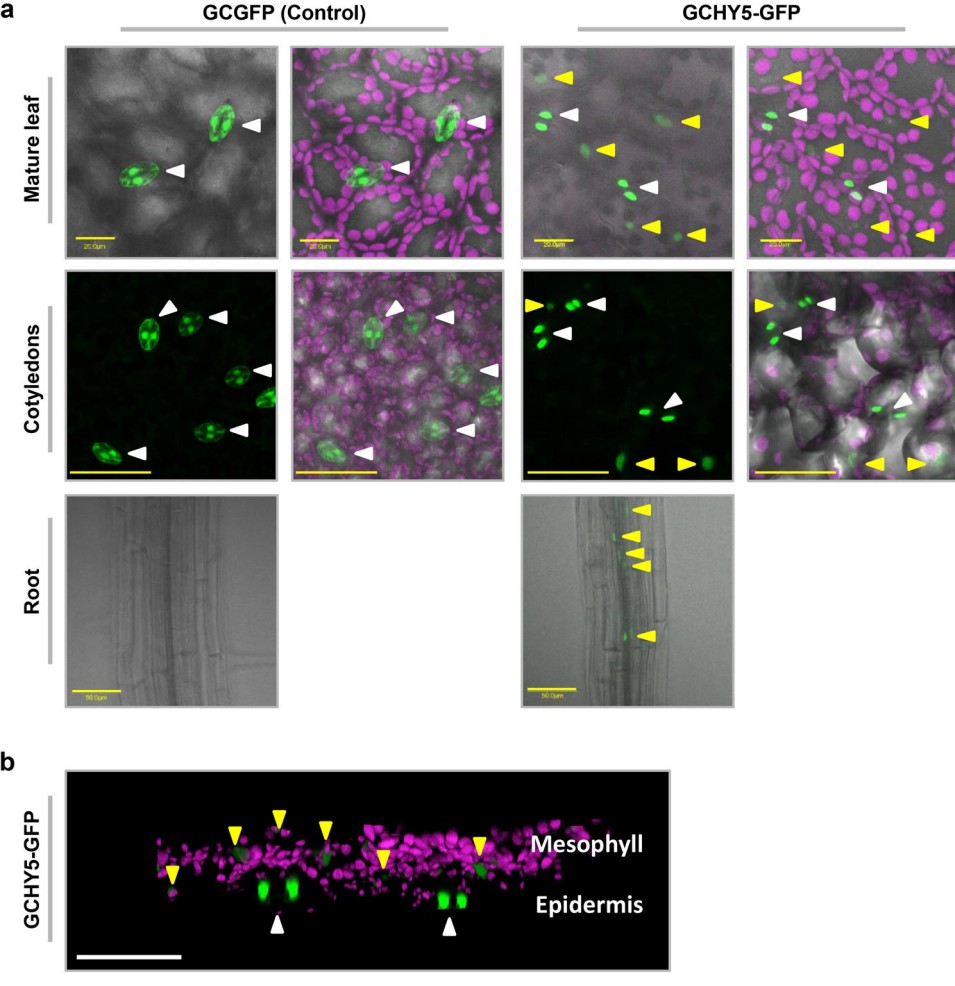

**Fig. 7 HY5 produced within guard cells is translocated to mesophyll and root cells. a** Distribution of GCGFP (control) or GCHY5-GFP in mature leaves, cotyledons and roots. All panels are merged images of white-light, chlorophyll-autofluorescence (stained magenta), and GFP-fluorescence (stained green). Scale bars (yellow) are 20 μm for mature leaves and 50 μm for cotyledons and roots. **b** 3D simulation providing side views of GCHY5-GFP cotyledons, composed of epidermis and mesophyll cell layers. Image is a merge of chlorophyll-autofluorescence (stained magenta), and GFP-fluorescence (stained green). Bar = 50 μm. **a, b** White arrows indicate the location of GFP in guard cells and yellow arrows indicate the location of GFP in mesophyll cells of mature leaves and the cotyledons, and within the phloem of the roots.

GCHXK/GCHY5/WT (WT background) was twice as high as that observed in the GCHXK/GCHY5/*hy5* or GCHY5/*hy5* lines, in which the expression of *HY5* was solely derived from GCHY5 (Fig. 6b). These results support the notion that *HY5* expressed in guard cells activates the expression of the endogenous *HY5*. Unlike *HY5*, the expression of *HXK1* derived from GCHXK was not affected in any of the crosses (Fig. 6c).

Since GCHXK per se promotes hypocotyl elongation against the WT background (Fig. 2), it could be that GCHXK suppresses the export of the endogenous HY5 from guard cells and, as a result, promotes hypocotyl elongation. To examine this possibility, we made plants that co-expressed *HXK1* and *HY5-GFP* in their guard cells (GCHXK/GCHY5-GFP plants) by crossing GCHXK2 with GCHY5-GFP and monitored GFP fluorescence (Fig. 8e–h). GFP fluorescence appeared in both the guard cells and mesophyll cells of the GCHXK/GCHY5-GFP, demonstrating that GCHXK does not block the export of HY5 from guard cells (Fig. 8f). Furthermore, the hypocotyls of GCHXK/GCHY5-GFP were short (Fig. 8h), similar to those of GCHXK/GCHY5/WT (Fig. 6a). This indicates that the fusion HY5-GFP retains its natural activity and inhibits the elongation imposed by GCHXK, most likely by auto-activating the expression of the endogenous *HY5*.

## Discussion
Germinating seedlings elongate in a search for light, which is perceived by photoreceptors and halts hypocotyl elongation. The perception of light prevents the degradation of HY5 and targets PIFs for degradation, thereby blocking hypocotyl elongation. Researchers have long wondered in which tissues and cell types the light signal is perceived. In the current study, we found that modulated expression of the sugar sensor *HXK1*, *HY5*, and *PIF4* in guard cells is sufficient to promote or stop hypocotyl elongation under light conditions, uncovering a role for guard cells in hypocotyl elongation. A recent study reported that expression of *PHYB* in the epidermis under the ML1 promoter is sufficient for the perception of a red-light signal and the inhibition of hypocotyl elongation in the *phyB* mutant[29]. Yet, the epidermis is composed of epidermal pavement cells and guard cells, with the latter appearing on the cotyledons immediately after germination[4,5]. Since the ML1 promoter is also active in guard cells ([56]; http://bar.utoronto.ca/efp/cgi-bin/efpWeb.cgi), it is possible that the light perception and the effects of *PHYB* that have been attributed to the epidermis are generated in the guard cells.

Many of the genes involved in hypocotyl elongation, including *PHYB*, *CRY1*, *CRY2*, *COP1*, *SPA*, *DET1*, *HY5*, *PIFs*, *YUC8*, and

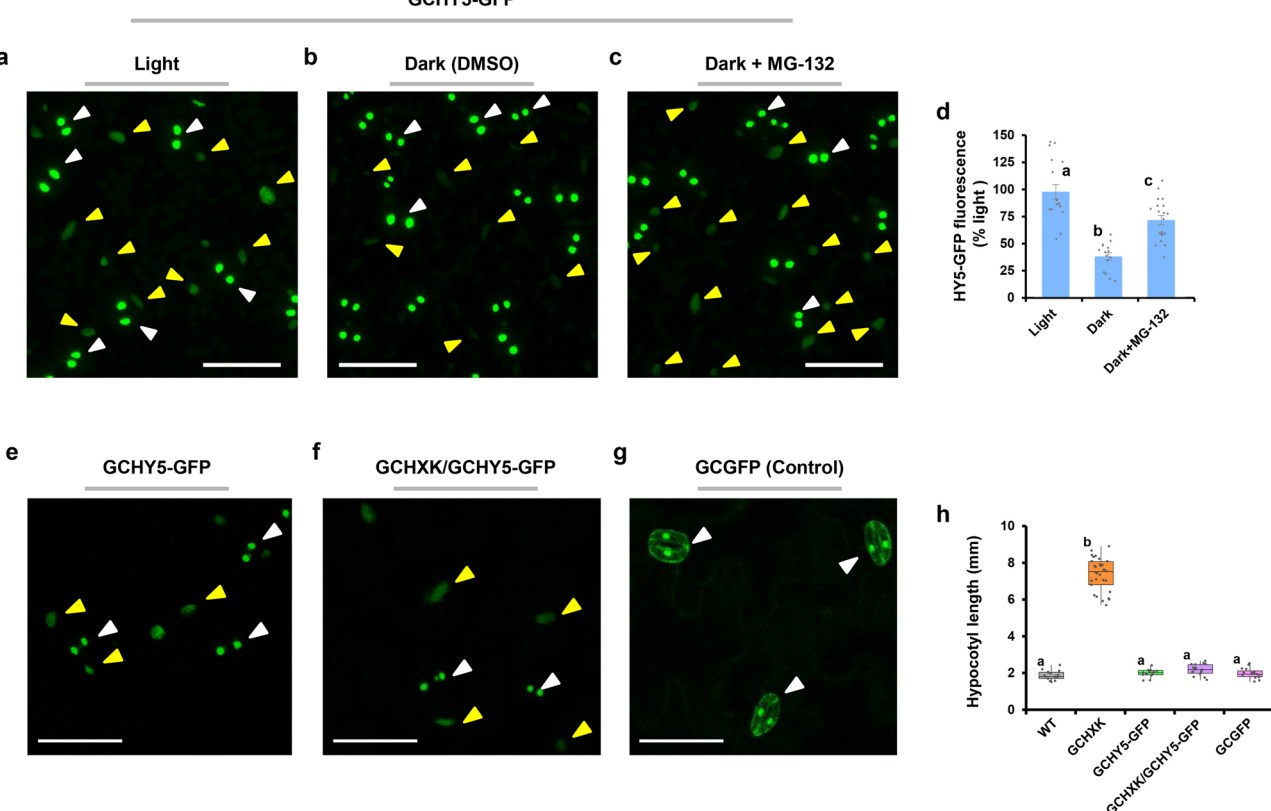

**Fig. 8 Dark and GCHXK do not block the export of HY5 from guard cells. a–d** HY5-GFP-fluorescence intensity is light-dependent. Ten-day-old GCHY5-GFP seedlings treated with 15 µM proteasome inhibitor; MG-132 **c**, kept in the dark for 16 h prior to image acquisition. **b** Dark-grown seedlings treated with 0.1% DMSO served as a control, and (**a**) light-grown seedlings served as an additional control. **d** Relative fluorescence intensity of HY5-GFP. The fluorescence intensity in the light was set to 100%. Data points are means ± SE ($n > 15$). Light gray dots represent individual data points. Different letters indicate a significant difference (Tukey's HSD test, $P < 0.05$). **e–h** GCHXK did not block the export of HY5 from guard cells. **e–g** Distribution of GFP signal in GCHY5-GFP (**e**), GCHXK/GFHY5-GFP (**f**), and GCGFP (**g**, control) in cotyledons of developing seedlings. **h** Hypocotyl lengths of GCHXK/GFHY5-GFP seedlings grown with 1% sucrose. The box plots extend from the first to third quartiles and the whiskers extend from the minimum to the maximum levels. Lines within the boxes signify median values and dots represent individual data points ($n > 15$). Different letters indicate a significant difference (Tukey's HSD test, $P < 0.05$). **a–c**, **e–g** All panels are GFP-fluorescence (stained green) images. White arrows indicate the location of GFP in guard cells and yellow arrows indicate the location of GFP in mesophyll cells. Bar = 50 µm (**a–c**), 25 µm (**e–g**).

*HXK*, are expressed in guard cells[57] (http://bar.utoronto.ca/efp/cgi-bin/efpWeb.cgi). We propose that light-activated photoreceptors within guard cells prevent the degradation of guard cells' HY5. HY5 then interferes with PIF4 activity to prevent the production of auxin within the guard cells[9–11]. In addition, HY5 exits the guard cells and apparently auto-activates *HY5* expression in other tissues, thereby interfering with the activity of PIF4 (and perhaps other PIFs, as well) in those tissues, and blocking hypocotyl elongation. Under dark conditions, HY5 that exits the guard cells is targeted for degradation by the proteosome, as the GCHY5-GFP signal in mesophyll cells of seedlings germinating in the dark was less than half of that of seedlings germinating in the light and the degradation was attenuated by a proteasome inhibitor (Fig. 8d).

Expression of *HY5* in guard cells is sufficient to partially inhibit hypocotyl elongation of the *hy5* mutant (Fig. 5c). It was previously shown that a chloroplast retrograde signal (sent from the chloroplast to the nucleus) is required for HY5-mediated repression of hypocotyl elongation[17]. It was also reported that the absence of functional chloroplasts prevents normal light perception and signal transduction by phytochromes[58]. While guard cells contain active chloroplasts[59], the presence of chloroplasts in the epidermal pavement cells has been debated[60]. In the course of this study, we failed to spot any chloroplast fluorescence

(autofluorescence of chlorophyll) in the pavement cells of epidermal peels, while chloroplast fluorescence was easily spotted in guard cells (Supplementary Fig. 5). It is likely that epidermal pavement cells in Arabidopsis cotyledons lack active chloroplasts and that guard cells rather than epidermal pavement cells mediate the de-elongation signal. Whether epidermal cells are also involved in the de-elongation signal should be examined using epidermal-specific promoters that do not drive expression in guard cells.

It has previously been shown that shade (a low ratio of red/far red light) is perceived in the cotyledons, where PIF4 drives the synthesis of the growth hormone auxin. Auxin is then transported to the hypocotyl to induce hypocotyl elongation[22,23,25]. Since hypocotyl elongation driven by *HXK1* expressed in guard cells (GCHXK) requires PIF4 (Fig. 4c) and since increased expression of *PIF4* in guard cells (GCPIF4) is sufficient to promote hypocotyl elongation in WT and *pif4* backgrounds even under long-day conditions (Fig. 5a, b), it is likely that GCHXK and GCPIF4 stimulate auxin production in the guard cells. The auxin is then transported to the epidermis of the hypocotyl, in which BR and gibberellin (GA) signals stimulate an elongation response[22,23,25]. Indeed, a higher level of auxin was measured in GCHXK seedlings (Fig. 3d), and the expression of PIF and auxin-induced genes were upregulated (Fig. 3a, b). In addition, inhibitor

of auxin transport prevented GCHXK-mediated hypocotyl elongation (Fig. 3e), supporting our assumption that GCHXK stimulates PIF4-mediated auxin production in guard cells.

The levels of hypocotyl elongation in the light of GCPIF4 seedlings with the *pif4*-mutant or WT backgrounds seem to be similar to one another and moderate (~30%; Fig. 5a, b). Unlike the expression of *PIF4* in guard cells, global over-expression of *PIF4* under 35S in the WT background stimulated much greater (doubled) hypocotyl elongation[61,62], indicating an additive elongation effect when *PIF4* expression is not limited to the guard cells. This conclusion is in line with our results; the more pronounced hypocotyl elongation of GCHXK (about 3 times greater than the WT) as compared to the moderate hypo-cotyl elongation of GCPIF4 (in which *PIF4* was overexpressed only in guard cells) is probably due to the expression of *PIF4* in GCHXK tissues other than guard cells. That is, GCHXK stimulates PIF4 activity not only in guard cells, but probably in other tissues as well. This assumption is further supported by the increased expression of *PIF1, 3* and *4* in GCHXK seedlings (Fig. 3a). That may suggest that expression of *PIF4* in the hypocotyl elongating tissues is also important for hypocotyl elongation[63]. The question of whether expression of any of the other PIFs, aside from *PIF4*, in guard cells and in the hypocotyl tissues is also required for elongation awaits further examination. Yet, the use of *PIF4* in this study shows that GCHXK stimulates hypocotyl elongation via the known COP1/SPA, DET1, and PIF pathways.

Unlike PIF4, the results from GCHY5 suggest that HY5 exits the guard cells and not only activates its expression in other tissues, but also prevents PIF-stimulated elongation effects in those tissues. This is based on the marked difference between GCHY5 expressed concomitantly with GCHXK in the WT background, as compared to the *hy5*-mutant background (Fig. 6). Expression of GCHXK and GCHY5 in the WT background completely abolished the elongation induced by GCHXK; whereas hypocotyl elongation was not abolished when GCHXK and GCHY5 were expressed in the *hy5*-mutant background (Fig. 6a). Since HY5 antagonizes the effect of PIFs, which are essential for hypocotyl elongation[12], these results indicate that auto-activated expression of *HY5* in locations other than the guard cells helps to eliminate PIF-stimulated elongation effect.

Our finding that HY5 moves from guard cells to other tissues (Fig. 7) is in line with the results reported by Chen[51], who demonstrated shoot-to-root HY5 movement. It has recently been shown that local expression of N-terminal GFP-tagged HY5 in epidermal, mesophyll or vascular tissues is confined to the specific tissue, with no movement of the tagged-HY5 out of the specific tissue[64]. Yet, this local expression was sufficient to partially (but not completely) shorten the long hypocotyls of the *hy5* mutant[64]. The authors of that work suggested that these results support the notion that HY5 acts in various tissues. Indeed, both that study and the current study suggest that various tissues are likely targets for autoactivation of *HY5* expression and participate in the inhibition of hypocotyl elongation by HY5. It is worth noting that in Burko's et al.[64] study, epidermal expression of the confined tagged-HY5 was driven by the CER6 promoter that also drives expression in the guard cells[65,66]. Yet, in our study, *HY5* expressed in guard cells of GCHY5 was not confined and moved out of the guard cells, which may explain why in some of the GCHY5/*hy5* lines (background of *hy5* mutant), the suppression of hypocotyl elongation was as complete as that observed in the WT (Fig. 5c).

The very fact that sugars promote hypocotyl elongation of WT Arabidopsis seedlings under short-[32] and long-day conditions (this study) suggests that light is not the only signal that controls hypocotyl elongation and that sugars can overcome the effects of light on this process. Furthermore, expression of *HXK1*, an established sugar sensor, under the global 35S or guard-cell promoter (35SHXK and GCHXK, respectively) stimulates hypocotyl elongation in the light, which also supports the notion that sugars constitute an additional pathway that affects hypocotyl elongation.

The observed effect of sucrose on the hypocotyl elongation of *hy5* mutants also supports the hypothesis that sugars constitute an independent regulatory pathway for hypocotyl elongation. The hypocotyls of *hy5* mutants elongate in the light, but the addition of sucrose causes it to elongate significantly further (Fig. 4a). That indicates an additive effect of sucrose over the absence of *hy5*, suggesting that sucrose does not require suppression of HY5 to promote hypocotyl elongation. Unlike the effect of sucrose on *hy5* mutants, sucrose added to *cop1* mutants does not promote hypocotyl elongation, suggesting that the effect of sucrose and HXK1 is dependent on COP1. COP1 is an E3 ubiquitin ligase which in darkness targets various transcription factors that inhibit hypocotyl elongation for ubiquitination and degradation, including HY5, HY5 HOMOLOG (HYH), LONG AFTER FAR RED LIGHT (LAF1), LONG HYPOCOTYL IN FAR RED (HFR1), B-box zinc finger proteins (BBXs), and a BR-regulated GATA transcriptional factor (GATA2)[6,49]. The fact that sucrose stimulates elongation beyond that of *hy5* mutant, but requires COP1 to stimulate elongation might indicate that sucrose stimulates elongation via the other targets of COP1, independent of HY5. Yet, it is still possible that sucrose also targets HY5 for degradation via COP1. The application of sugar represses the transcription of *HY5* and its homolog *HYH*, as seen in a database search (Supplementary Fig. 6). In line with those results, we found that GCHXK reduced the expression of *HY5* (Figs. 3a, 6b). Therefore, it is possible that sucrose and HXK1 promote hypocotyl elongation via several mechanisms: inducing expression of *DET1* (Fig. 3c) helping DET1 and COP1 to target various hypocotyl-elongation inhibitors for degradation (including HY5), repressing the expression of *HY5* (perhaps by reducing auto-activation of *HY5* expression) and possibly inducing the expression of PIF genes independent of HY5 (Fig. 3a).

The results of the current study suggest that the effect of guard cells on hypocotyl elongation involves two opposite signals converging at PIF4[33]. The first is sugar-sensing by HXK1, which overcomes the effect of the endogenous HY5 to promote hypocotyl elongation. This effect of HXK1 is dependent on COP1, DET1, and PIF4 (and perhaps other PIFs) and involves the generation of an auxin signal that exits the guard cells and activates elongation effects in the target tissues. The second signal is perceived by photoreceptors, probably within guard and meso-phyll cells[29,67], prevents degradation of HY5 and blocks PIF activity. It is likely that there is some balance between HXK1 and HY5 signals within guard cells. HXK1 tilts the balance toward hypocotyl elongation, while HY5 tilts the balance towards movement and autoactivation of *HY5* expression in other tissues, which eventually inhibit hypocotyl elongation.

We have shown that GCHXK overcomes the inhibition of elongation caused by blue light, but not that caused by red light (Fig. 2f, g). We suggest the following possible explanation for this difference: At dawn, the primary light is blue light, which is photosynthetically inefficient and, therefore, the primary carbon source at that time of day is sucrose generated from seed storage. As the sun rises, the blue light is followed by photosynthetically efficient red light, with which the immediate products of photo-synthesis are triose phosphates. While triose phosphates are phosphorylated independently of HXK activity, sucrose metabo-lism requires hexose phosphorylation of the sucrose-cleavage products, glucose and fructose, by HXK. We, therefore, suggest that HXK activity acts as a sensor for photosynthetic efficiency.

That is, HXK indicates how much of the available carbon is derived from storage reserves versus photosynthesis-derived triose phosphates. Under blue-light conditions, a high level of HXK activity, which is needed for sucrose metabolism, indicates that the carbon is coming from storage reserves, which promotes hypocotyl elongation in a search for photosynthetically efficient red light. However, under red-light conditions associated with the production of triose phosphates, HXK activity is dispensable, and therefore, no further elongation is promoted.

In summary, the current study reveals that guard cells are central players in hypocotyl elongation. It also reveals the role of HXK1 in that elongation and indicates that light and sucrose antagonistically coordinate the effort to achieve the height necessary for efficient photosynthetic, autotrophic sugar production.

## Methods

**Plant material**. All of the plants used in this study were of the *Arabidopsis thaliana* Col-0 ecotype. The *hy5* mutant was obtained from the Arabidopsis Information Resource (ABRC) stock (Salk_096651, https://abrc.osu.edu). The *pif4-101*, *cop1-4*, *pif3-5*[68], *det1*[69], GCHXK, GCGFP[44,46], *35SHXK*[47], and *35SFRK1*[48] plants have been described previously. Cloning, transformation and characterization of additional transgenic lines, as well as crosses and T-DNA mutants used in this study, are described below.

**Plasmid construction**. To generate the $KST_{ppro}$::AtPIF4 (GCPIF4), $KST_{ppro}$::AtHY5 (GCHY5), $35S_{pro}$::AtHY5 (35SHY5) and $KST_{ppro}$::AtHY5-GFP (GCHY5-GFP) transgenic plants, segments for the coding regions of *AtHY5* (AT5G11260), *AtHY5* fused to GFP (HY5-GFP) and *AtPIF4* (AT2G43010), as well as a segment of the *KST1* partial promoter ($KST_{ppro}$[5]) were synthesized by GENEWIZ (https://www.genewiz.com). Specific restriction sites (*Not*I, *Xho*I, *Xma*I and *Xba*I) were added to the sequence. The binary vector *p*Green containing an insertion of "NotI-35Spro-XhoI-XmaI-GFP-XbaI" that was kindly provided by the lab of Dr. Arthur Schaffer (ARO, Israel) was used to clone the GCPIF4, GCHY5, 35SHY5, and GCHY5-GFP constructs. For the generation of the *35SFRK2* construct, the *Solanum tuberosum* Fructokinase2 (GeneBank accession number AF106068) was inserted under the 35S promoter in the binary vector pBI121[70].

**Generation of transgenic plants**. Electrocompetent *Agrobacterium tumefaciens* (GV3101 strain) was transformed using 100 ng plasmid by electroporation. *Arabidopsis* WT, *hy5*, and *pif4* plants were transformed using the floral-dip method[71]. Screening was performed on 0.5× MS selection media containing 3% sucrose (Suc, Duchefa) and 50 mg l$^{-1}$ kanamycin (Kan; Duchefa). The primers used to identify positive transgenic events are listed in Supplementary Table 3. T-DNA single mutant lines were genotyped by primers designed using the SIGnAL primer design tool (http://signal.salk.edu/tdnaprimers.2.html) powered by the Genome Express Browser Server (GEBD). Homozygous mutants were identified using the primers listed in Supplementary Table 2. GCHXK crosses conducted in this study were all carried out using the same GCHXK line (GCHXK2). For the GCHXK/GCHY5, GCHXK/GCHY5/*hy5* crosses, we used lines GCHY5/*hy5* #12 (described in Fig. 5c) together with GCHXK2. The primers used for the identification of the crosses, GCHXK/*pif4*, GCHXK/*hy5*, GCHXK/*det1*, GCHXK/GCHY5, GCHXK/GCHY5/*hy5* and GCHXK/GCHY5-GFP, are listed in Supplementary Tables 2, 3.

**Hypocotyl-growth assay**. For the hypocotyl-growth assay, seeds were sterilized and sown onto Petri plates (9 mm diameter) containing half Murashige and Skoog (0.5× MS; Duchefa Biochemie, The Netherlands) medium as a control (pH 5.8, 0.8% plant agar) or 0.5× MS with varying sugar concentrations; 1, 2, or 3% sucrose (Suc; Duchefa), as indicated for each experiment. Seeds were cold-treated at 4 °C in darkness for 3 days before they were transferred to a growth chamber (16 h light/8 h dark photoperiod at 22–23 °C, 40 μmol m$^{-2}$ s$^{-1}$ light intensity). For plants grown in soil, a potting mix containing (w/w) 70% peat, 30% perlite, supplemented with slow-release fertilizer (Even-Ari, Israel) was used. For seedlings grown under blue and red light conditions, we used the adjustable led lighting system, pro 325 (Lumigrow, CA, USA). Seedlings were imaged 7 days after the transfer to the growth room, unless mentioned otherwise. To determine hypocotyl length, images were analyzed using the ImageJ software (http://rsb.info.nih.gov/ij/) fit-line tool, following size calibration.

**Scanning electron microscopy**. For the scanning electron microscopy (SEM), seedlings were fixed in 3.7% formaldehyde, 50% ethanol, and 5% acetic acid by vacuum infiltration for 30 min. Seedlings were later kept in the fixative for 8 h, followed by a slow dehydration through a series of ethanol concentrations: 50%, 70% and 90%, 100%, 100%, 60 min each. Seedlings were critical-point dried in

liquid $CO_2$ in a Quorum K850 critical-point dryer (Quorum Technologies, East Sussex, UK), and sputter-coated with gold palladium using a Quorum SC7620 mini sputter coater (Quorum Technologies, East Sussex, UK). Images were taken with a JEOL JCM6000 benchtop SEM (Jeol, Japan).

**Microarray database processing and analysis**. Expression data for *AtHY5* and *AtHYH* were obtained from the NASCarrays microarray database[72], Experiment No. 593. A heat-map diagram was computed using Expander 7 software[73] based on the expression data obtained from the NASCArray database. The accession number and the Affimetrix probe ID number for each gene are listed in Supplementary Table 4. Gene accession numbers were assigned according to the information in the TAIR database (http://www.arabidopsis.org/).

**RNA extraction and cDNA preparation**. Samples were collected from seedlings grown on 0.5× MS agar plates, 4 h after lights were switched on. Each sample included at least 40 seedlings. Samples were snap-frozen in liquid nitrogen and total RNA was extracted using the Logspin method[74]. Samples were ground using a Geno/grinder (SPEX SamplePrep, Metuchen, NJ, USA) and RNA was extracted in 8M guanidine hydrochloride buffer (Duchefa Biochemie) and then transferred to tubes containing 96% EtOH (Bio Lab, Jerusalem, Israel). Then, samples were transferred through a plasmid DNA extraction column (RBC Bioscience, New Taipei City, Taiwan), washed twice with 3M Na-acetate (BDH Chemicals, Mumbai, India) and twice in 75% EtOH and eluted with DEPC (diethylpyrocarbonate) water (Biological Industries, Co., Beit Haemek, Israel) that had been preheated to 65 °C. The RNA was treated with RQ1-DNase (ProMega, Madison, WI, USA) according to the manufacturer's instructions, to degrade any residual DNA. The purity of all RNA samples was assessed by 260/280 and 260/230 nm absorbance ratios. For the preparation of cDNA, total RNA (1 μg) was taken for reverse-transcription PCR using qscript$^{TM}$ cDNA Synthesis Kit (Quanta BioSciences, Gaithersburg, MD, USA) following the manufacturer's instructions.

**Quantitative real-time PCR**. For qPCR, cDNA samples were diluted 1:7 in DEPC water. Quantitative real-time PCR reactions were performed using SYBR Green mix (Thermo-Scientific, Waltham, MA, USA) and reactions were run in a RotorGene 6000 cycler (Corbett, Mortlake, New South Wales, Australia). Following an initial pre-heating step at 95 °C for 15 min, there were 40 cycles of amplification consisting of 10 s at 95 °C, 15 s at 55 °C, 10 s at 60 °C, and 20 s at 72 °C. The melting point was determined for each sample to validate the specificity of the primers. Results were analyzed using the RotorGene software. The Arabidopsis *AtTUB2* (accession no. At5g62690) was used as a reference for the normalization of cDNA amounts. The primers used for the amplification procedure are listed in Supplementary Table 1.

**Library preparation and sequencing**. Library construction and sequencing were performed by the Genomics Unit at the Grand Israel National Center for Personalized Medicine (https://g-incpm.weizmann.ac.il), Weizmann Institute of Science (Rehovot, Israel). Briefly, the poly(A) fraction (mRNA) was purified from 500 ng of total RNA, followed by fragmentation and generation of double-stranded cDNA. Then end repair, A-base addition, adaptor ligation and PCR amplification were carried out. Sequencing libraries were constructed with barcodes to allow the multiplexing of eight samples in one lane. An Illumina HiSeq 2500 V4 instrument was used to sequence single-end, non-stranded 60-bp reads. The number of reads was similar for all samples.

**Filtering and mapping of reads to the reference genome**. Raw reads were subjected to a filtering and cleaning procedure. Adaptors were removed using Trimmomatic software, version 0.32[75]. Then, the FASTX Toolkit (http://hannonlab.cshl.edu/fastx_toolkit/index.html, version 0.0.13.2) was used to (i) trim read-end nucleotides with quality scores <30, with the Fastq quality_trimmer, and (ii) remove reads with <70% of base pairs with a quality score ≤30 using the Fastq quality filter. Transcript quantification (the number of reads per gene) from the RNA-Seq data was performed using the Bowtie2 aligner[76], the Arabidopsis reference genome (TAIR database) and the expectation-maximization method (RSEM) to estimate maximum-likelihood expression levels[77]. The expression level was calculated as trimmed mean of M values (TMM)-normalized counts[78].

**Differential-expression analysis, GO analysis, and pathway enrichment**. Differential-expression (DE) analysis was performed with the DESeq2 package[79] of the R software. Transcripts that were more than 2-fold differentially expressed with an adjusted *P* value of no more than 0.05 were considered differentially expressed. GO and pathway-enrichment analyses were performed using KOBAS program[80], to determine significant annotated processes of the main biological functions. The PANTHER (http://www.pantherdb.org), KEGG PATHWAY (www.genome.jp/kegg/) and BioCyc (https://biocyc.org) Gene Ontology databases were used with multiple testing correction of false discovery rate (FDR)[81]. The threshold was set to a FDR with a corrected *P* value of <0.05. The REVIGO web server[82] was used for removing redundant GO terms. DE transcript diagrams were displayed using

MapMan 3.6.0RC1 (https://mapman.gabipd.org/mapman-version-3.6.0) and TAIR10 mapping (Ath_AGI_TAIR9_Jan2010).

**Confocal microscopy imaging**. Image acquisition was done using a Leica SP8 laser scanning microscope (Leica, Wetzlar, Germany) equipped with a solid-state laser with 488 nm light, a HC PL APO CS 63×/1.2 water immersion objective (Leica) and Leica Application Suite X software (LASX, Leica). Images of GFP signal were acquired using the 488-nm laser light and the emission was detected with HyD (hybrid) detector in a range of 500–525 nm. Autofluorescence of the chloroplasts was detected in a range of 650–750 nm with a PMT detector.

**Auxin quantification using LC–MS**. Hormone extraction was performed using standard protocols[83] with slight modifications. Five-day-old seedlings grown on control agar plates (0.5× MS, 0.8% plant agar, pH 5.8) were collected and frozen in liquid nitrogen. Each sample collected, included at least 80 seedlings. The frozen tissue was ground to a fine powder using a Geno/grinder (SPEX SamplePrep, Metuchen, NJ, USA). Two hundred mg of the powder were transferred to a 1.5-ml tube containing 1 ml of an extraction solvent (ES) mixture [79% isopropanol (Bio Lab, Israel): 20% MeOH (Bio Lab): 1% acetic acid, (Gadot, Israel)], supplemented with 20 ng of each deuterium-labeled internal standards (IS, Olomouc, Czech Republic). The tubes were incubated for 60 min at 4 °C with rapid shaking and centrifuged at 14,000 g for 15 min at 4 °C. The supernatant was collected and transferred to a 2 ml tube. ES (0.5 ml) was added to the pellet and the extraction steps were repeated twice. The combined extracts were evaporated using a speed-vac (Hetovac VR-1, Denmark) at room temperature. Dried samples were dissolved in 200 μl of 50% methanol and filtered through a 0.22 μm PVDF syringe filter (Agela Technologies, Torrance, CA, USA). Five μl were injected for each analysis.

LC–MS-MS analyses were conducted using UPLC-Triple Quadrupole-MS (Waters Xevo TQ MS). Separation was performed on a Waters Acquity UPLC BEH C18 1.7 μm 2.1 × 100 mm column with a VanGuard pre-column (BEH C18 1.7 μm 2.1 × 5 mm). Chromatographic parameters were as follows: The mobile phase consisted of water (phase A) and acetonitrile (phase B), both containing 0.1% formic acid in the gradient-elution mode. The solvent gradient program for auxins is presented in Supplementary Table 6. The flow rate was 0.3 ml/min and the column temperature was kept at 35 °C. The retention times and MS-MS parameters for each plant hormone and its internal standard are listed in Supplementary Table 5. Acquisition of LC–MS data was performed using MassLynx V4.1 software (Waters). Quantification was done using isotope-labeled IS, except for OxIAA, which was quantified using calibration curves.

**Auxin transport activity assay**. For the auxin transport activity assay, the auxin transport inhibitor N-1-naphthylphthalamic acid (NPA, Sigma-Aldrich, Israel) dissolved in dimethyl sulfide (DMSO, Sigma-Aldrich, Israel) was added to the 0.5× MS agar media, containing 1% Suc. A 0.5× MS + 1% Suc media, supplemented with 0.1% DMSO without NPA, served as the control. Hypocotyl length was measured at 9 days after germination.

**Proteasome inhibitor assay**. For the proteasome inhibitor treatment, we used carbobenzoxy-Leu-Leu-leucinal (MG-132, Sigma-Aldrich, Israel) dissolved in DMSO. Ten-day-old GCHY5-GFP seedlings grown on 0.5x MS agar media containing 1% Suc (grown under long-day conditions) were treated with MG-132 to a final concentration of 15 μM. Following the application of MG-132, seedlings were moved to the dark for 16 h. DMSO (0.1%) served as a control. Following MG-132/dark treatment, cotyledons were taken for confocal imaging. Images were analyzed using the ImageJ software histogram tool to evaluate fluorescence intensity.

**Calculated lengths of hypocotyl epidermal cells**. The average cell length of 4-day-old seedlings was calculated by dividing hypocotyl length by the average number of cells counted following the SEM analysis.

**Statistical analysis**. Statistical analysis was performed using the JMP 14 software program. Box plots were prepared using the Graph Builder tool. Means were compared using Student's t test, Tukey's HSD test or Dunnett's method, as described for each experiment. Means were considered to be significantly different at $P < 0.05$.

**Reporting summary**. Further information on research design is available in the Nature Research Reporting Summary linked to this article.

## Data availability

The sequencing data from this study have been deposited in the The National Center for Biotechnology Information BioProject database (https://www.ncbi.nlm.nih.gov/bioproject/) under the accession number PRJNA687355. The raw data from the Illumina sequencing have been deposited in NCBI Sequence Read Archive (SRA; https://www.ncbi.nlm.nih.gov/sra) under accession numbers SRR13316969-SRR13316975. The raw data referring to the plots shown in the figures are provided in Supplementary Data 1, 2. All relevant data are available from the authors upon request.

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

## Acknowledgements

We would like to thank the reviewers for their valuable comments that have improved the paper. We thank Prof. Jorge Casal and Manuel Pacín for providing the *pif4-101, cop1-4, pif3-5* seeds, Prof. Meng Chen for the *det1* seeds and Prof. Danny Chamovitz for the *phyB* seeds. We thank Dr. Mira Carmeli-Weissberg (Volcani Center, ARO) for assisting us with the LC–MS-MS analysis. We would also like to thank Dr. Joshua Klein for providing the LED lighting systems.

## Author contributions

G.K., N.C., and D.G. planned and designed the research. G.K. carried out plasmid construction and generation of transgenic plants. G.K., O.S., N.L., and A.E. generated, genotyped, and analysed the crosses presented in this study. G.K., A.E., N.S., and D.B. performed the experiments, analysed the data, and interpreted the results. A.D.-F. carried out bioinformatics analysis and interpretation. E.B. carried out confocal microscopy imaging, F.S and G.K. conducted auxin quantification, and H.Z., D.B., and N.L. performed SEM experiments. G.K., N.S., and D.G. wrote the paper.

## Competing interests

The authors declare no competing interests.
