## [Peer Review File · Communications Biology]

Reviewers' comments:

Reviewer #1 (Remarks to the Author):

Manuscript by Kelly et al., highlights an very important and novel finding in linking sucrose and growth specifically hypocotyl growth via guard cell derived signals. It is interesting that while light inhibits elongation, the by-product of light signalling i.e accumulation of sucrose has an opposing effect. The authors in this study have very elegantly shown how specific cell type i.e guard cells integrate light signals to control hypocotyl growth by cell specific expression of HXKs and transcription factors such as HY5 and PIF4. Overall, the study is novel, and well designed with proper controls. However, the study has number of concerns, which are essential in understanding the COP1/DET1-HY5/PIFs signaling module in GC mediated hypocotyl elongation

1. In Fig1. can authors provide the possible mechanism through which HXKs promote hypocotyl elongation ? Is GCHXK mediated hypocotyl elongation is specific to any particular wavelength or independent of wavelength?
2. The authors should show better seedling pictures in Fig2A. as the one that is provided is not clear. In Fig 2d, it would be nice to show quantifying data for the cell length for different genotypes.
3. In Fig4, is the GC mediated hypocotyl elongation is PIF4 specific or it requires other PIFs such as PIF3 and PIF5. I think authors should check hypocotyl response for other pif mutants including pifQ. Also, it will be nice to show response curve for all the mutants.
4. In fig 5. is GCHXK mediated promotion of hypocotyl elongation is specific to pif4i only or is it that it requires other PIFs also. Authors should at least test this using pif3 and pif5 mutants
5. In Fig 6, it's quite confusing to show WT, pif and hy5 mutants separately. Is it possible to redo this and show in the same graph, this would help the reader to make better and direct comparison.
6. In Fig 7, is it possible that any of the HXKs expression is regulated by HY5 in the guard cells? Can authors provide some evidence in this direction?
7. In Fig 11, on what basis the authors show arrow from HXKs to COP1 and DET1? Is there any evidence to suggest that sugar signalling promotes COP1/DET1 activity?
8. Is there any effect of sucrose on the stability of HY5, PIF4, PHYB, COP1 and DET1 proteins? This would help in understanding the mechanism of hypocotyl elongation to some extent?
9. What is the HY5 protein status in Guard cells in dark and light conditions with and without sucrose?

Reviewer #2 (Remarks to the Author):

This manuscript by Kelly et al. reports an interesting and important study on how hypocotyl growth of Arabidopsis seedlings is controlled at cellular level. Based on the observation that either constitutive or guard-cell-specific expression of HXK in plants results in similar elongated hypocotyls, the authors argue that guard-cell expressed HXK is sufficient to drive hypocotyl elongation. The further gene expression and genetic evidence basically support that HXK might promote hypocotyl grown through increasing the activity of PIF4 (the positive regulator of hypocotyl growth) and auxin level, and competing with the effects of HY5 (negative regulator of hypocotyl growth). Consistent with the reported mobile property of HY5 proteins, the authors showed that light signals broadened the cell-type distributions of guard-cell-expressed HY5 proteins to mesophyll cells. Overall, the manuscript is well written, and I find the idea in this study exciting. But I also have concerns about much of the genetic data as presented, and some suggestions for new experimental results.

Point 1. Some genetic materials used in Fig. 5 and 7 are not clearly explained. For example, is GCHXK/hy5 (in Fig. 5a) prepared by plant transformation or crossing GCHXK/WT with hy5 mutants? Unless the expression level or genomic location of GCHXK transgene remained unchanged in WT and hy5, the hypocotyl phenotype is not comparable between the single line of GCHXK/WT and GCHXK/hy5. Otherwise multiple independent transgenic lines in WT and hy5 should be used for phenotypic comparison. If this is the case, the expression level of transgene (e.g HXK) in different lines are also needed. The same problem is also seen for GCHXK/pif4 (in Fig.

5b), GCHXK/det1 (in Fig. 5c), GCHXK/GCHY5/WT and GCHXK/GCHY5/hy5 (in Fig. 7). Some key arguments made based on the observations using those materials might be jeopardized by inappropriate genetic analyses.

Point 2. Fig. 3b shows an increase of auxin level in GCHXK plants, and this result is critical to explain the phenotype of elongated hypocotyl caused by GCHXK expression. Therefore, it is necessary to see if inhibition of auxin activities in GCHXK plants reduces the hypocotyl elongation. For example, supplement growth medium with NPA to inhibit polar transport of auxin and see how it effects hypocotyl elongation of GCHXK seedlings.

Point 3. I find it difficult to understand the results shown in Fig. 7a. GCHY5 itself can restore the long-hypocotyl phenotype of hy5 to varying extents (Fig. 6c) and play dominant roles over GCHXK (Fig. 7a), it is hard to believe that GCHY5's activity in GCHXK/GCHY5/hy5 is strikingly restrained as presented in Fig. 7a. Did the authors examine the expression levels of GCHXK and GCHY5 in different genotypes used in this figure?

Point 4. The authors claimed that GCHY5-GFP proteins can be exported to mesophyll cells in the presence of light (Fig. 8-10). Since HY5 proteins are degraded in the dark and stabilized in the light, the steady state level of HY5 proteins would be expected much lower in the dark than in the light. Therefore, alternative interpretation of the results in Fig. 8-10 would be that a small portion of GCHY5-GFP proteins could move out of guard cells regardless of light, which are below the detection limit under microscope in the dark due to protein degradation. New experiment may be needed to clarify this point, for example, pre-treat the GCHY5-GFP seedlings with MG132 in the dark to block HY5 degradation, then examine the location of GCHY5-GFP proteins in the same condition.

MS ID#: COMMSBIO-20-2629A

MS Title: Guard cells control hypocotyl elongation through HXK, HY5 and PIF4

Response to the referees

Reviewers' comments:

Reviewer #1 (Remarks to the Author):

Manuscript by Kelly et al., highlights an very important and novel finding in linking sucrose and growth specifically hypocotyl growth via guard cell derived signals. It is interesting that while light inhibits elongation, the by-product of light signalling i.e accumulation of sucrose has an opposing effect. The authors in this study have very elegantly shown how specific cell type i.e guard cells integrate light signals to control hypocotyl growth by cell specific expression of HXKs and transcription factors such as HY5 and PIF4. Overall, the study is novel, and well designed with proper controls. However, the study has number of concerns, which are essential in understanding the COP1/DET1-HY5/PIFs signaling module in GC mediated hypocotyl elongation

Comment #1: In Fig1. can authors provide the possible mechanism through which HXKs promote hypocotyl elongation ? Is GCHXK mediated hypocotyl elongation is specific to any particular wavelength or independent of wavelength?

Response: In response to the reviewer's suggestion, we examined hypocotyl growth under different intensities of monochromatic blue and red light (Fig. S5). We found that GCHXK stimulates elongation under blue light, but not under red light. We describe these results in the Results section and discuss the biological meaning of the different effects of blue and red light on GCHXK elongation in the Discussion section. The immediate molecular mechanism by which HXK activates COP1/SPA and DET1 is yet to be discovered, but we suggest that it may involve the induction of the expression of these genes.

Comment #2: The authors should show better seedling pictures in Fig2A. as the one that is provided is not clear. In Fig 2d, it would be nice to show quantifying data for the cell length for different genotypes.

Response: The image in the original Figure 2A was removed. Cell-length data were added, as suggested by the reviewer (Fig. 2E).

Comment #3: In Fig4, is the GC mediated hypocotyl elongation is PIF4 specific or it requires other PIFs such as PIF3 and PIF5. I think authors should check hypocotyl response for other pif mutants including pifQ. Also, it will be nice to show response curve for all the mutants.

Response: No doubt this study could be expanded to include the examination of each of the PIFs individually, but the aim of this study was to show that HXK promotes elongation via the classical COP1, DET1, HY5 and PIF pathway. We, therefore, used PIF4, which is central to elongation. Nonetheless, following the reviewer's

suggestion, we added the results for the triple mutant *pif3-5* (Fig. 4A). We found that, as expected, the *pif3-5* hypocotyls were shorter than those of the WT and PIF4. Perhaps this is not surprising, because the triple mutant includes the *pif4* mutation. As for the response curve, we guess the reviewer is referring to the response to increasing sucrose levels. But since sucrose levels above 1% had no further elongation effect even on the WT (Fig. 1), it is very unlikely that they would stimulate hypocotyl elongation in the *pif* mutants.

Comment #4: In fig 5. is GCHXK mediated promotion of hypocotyl elongation is specific to pif4i only or is it that it requires other PIFs also. Authors should at least test this using pif3 and pif5 mutants

Response: Please see the response to Comment #3. GCHXK/*pif4* was generated by crossings of the *pif4* mutant with the well-characterized GCHXK2 line, and the results presented in this study were produced using T3 double-homozygous plants. The creation of such a T3 homozygote line takes more than one year to complete. We believe that the generation of GCHXK/*pif3* and GCHXK/*pif5* is not crucial for the central message of this study, namely, that the stimulation of hypocotyl elongation by HXK is mediated by COP1, DET1, HY5 and PIF. However, we do not claim that PIF4 is the only PIF involved, and know that further studies should examine the roles of each of the other individual PIFs in HXK-mediated elongation.

Comment #5: In Fig 6, it's quite confusing to show WT, pif and hy5 mutants separately. Is it possible to redo this and show in the same graph, this would help the reader to make better and direct comparison

Response: We added colors that better show the differences between the transgenes. Since we refer to each of the experiments independently in the text, we believe it is best to present them separately.

Comment #6: In Fig 7, is it possible that any of the HXKs expression is regulated by HY5 in the guard cells? Can authors provide some evidence in this direction?

Response: We added an expression analysis of HXK1 for the various lines assayed in this figure [GCHXK, GCHY5, and GCHXK/GCHY5, with either WT or *hy5* background (Fig. 6C)]. The expression of HXK1 is not affected by the addition of HY5 or by the lack of HY5. Rather, our results suggest that HXK suppresses the expression of HY5, as shown in GCHXK (Fig. 6B). Adding GCHY5 on top of GCHXK increases HY5 expression to higher levels than observed in WT, but does not affect HXK1 levels.

Comment #7: In Fig 11, on what basis the authors show arrow from HXKs to COP1 and DET1? Is there any evidence to suggest that sugar signalling promotes COP1/DET1 activity?

Response: This figure was removed.

Comment #8: Is there any effect of sucrose on the stability of HY5, PIF4, PHYB, COP1 and DET1 proteins? This would help in understanding the mechanism of hypocotyl elongation to some extent?

Response: It is quite complicated to distinguish between the effect of sucrose on the expression of any of these genes and its effect on the stability of the protein. We do see increased expression of *DET1* in *GCHXK* plants, accompanied by reduced expression of *HY5* and increased expression of *PIF1,3,5* (Fig. 2). As *COP1* and *DET1* are known to target *HY5* protein for degradation, we would expect to see changes in protein stability or protein levels. If the reviewer meant protein level instead of stability, then the data for protein level in response to Suc was addressed in earlier studies. For example, *PIF* and *HY5* protein levels were found to increase and decrease, respectively, in response to Suc¹⁻³, corroborating our expression results.

Comment #9: What is the HY5 protein status in Guard cells in dark and light conditions with and without sucrose?

Response: We show (Fig. 8) that *HY5* proteins produced in guard cells of seedlings grown in the presence of sucrose move out of the guard cells to mesophyll cells, where they are degraded in the dark, while the *HY5* signal within the guard cells remain high. Using a proteasome inhibitor, we confirmed that the *HY5* in the mesophyll under dark conditions is subject to ubiquitination, most probably by *COP1*.

Reviewer #2 (Remarks to the Author):

This manuscript by Kelly et al. reports an interesting and important study on how hypocotyl growth of Arabidopsis seedlings is controlled at cellular level. Based on the observation that either constitutive or guard-cell-specific expression of HXK in plants results in similar elongated hypocotyls, the authors argue that guard-cell expressed HXK is sufficient to drive hypocotyl elongation. The further gene expression and genetic evidence basically support that HXK might promote hypocotyl growth through increasing the activity of PIF4 (the positive regulator of hypocotyl growth) and auxin level, and competing with the effects of HY5 (negative regulator of hypocotyl growth). Consistent with the reported mobile property of HY5 proteins, the authors showed that light signals broadened the cell-type distributions of guard-cell-expressed HY5 proteins to mesophyll cells. Overall, the manuscript is well written, and I find the idea in this study exciting. But I also have concerns about much of the genetic data as presented, and some suggestions for new experimental results.

Comment #1: Some genetic materials used in Fig. 5 and 7 are not clearly explained. For example, is GCHXK/hy5 (in Fig. 5a) prepared by plant transformation or crossing GCHXK/WT with hy5 mutants? Unless the expression level or genomic location of GCHXK transgene remained unchanged in WT and hy5, the hypocotyl phenotype is not comparable between the single line of GCHXK/WT and GCHXK/hy5. Otherwise multiple independent transgenic lines in WT and hy5 should be used for phenotypic comparison. If this is the case, the expression level of transgene (e.g HXK) in different lines are also needed. The same problem is also seen

for *GCHXK/pif4* (in Fig. 5b), *GCHXK/det1* (in Fig. 5c), *GCHXK/GCHY5/WT* and *GCHXK/GCHY5/hy5* (in Fig. 7). Some key arguments made based on the observations using those materials might be jeopardized by inappropriate genetic analyses.

Response: We provide a detailed explanation as to how the genetic material was prepared. All lines containing *GCHXK* were created by crossing the very well-characterized *GCHXK2* line with the specific mutant. The *GCHXK* crosses (*GCHXK/hy5*, *GCHXK/pif4*, *GCHXK/det1*, *GCHXK/GCHY5/WT*, *GCHXK/GCHY5/hy5*, *GCHXK/GCHY5-GFP*) were all carried out using the same *GCHXK* line (line #*GCHXK2*). For the *GCHXK/GCHY5*, *GCHXK/GCHY5/hy5* crosses, we used lines *GCHY5/hy5* #12 (described in Fig. 5C), together with *GCHXK2*. This is now clearly specified in the Methods section ('Generation of transgenic plants' sub-section) and in the main text. Hence, no position effect with regard to *GCHXK* is expected and, therefore, there is no need to examine multiple independent transgenic lines.

Comment #2: Fig. 3b shows an increase of auxin level in *GCHXK* plants, and this result is critical to explain the phenotype of elongated hypocotyl caused by *GCHXK* expression. Therefore, it is necessary to see if inhibition of auxin activities in *GCHXK* plants reduces the hypocotyl elongation. For example, supplement growth medium with NPA to inhibit polar transport of auxin and see how it effects hypocotyl elongation of *GCHXK* seedlings.

Response: According to the reviewer suggestion, we carried out an experiment using NPA (Fig. 3E), which clearly showed that inhibiting auxin transport via the polar auxin transport inhibitor NPA prevented the elongation of the hypocotyls of *GCHXK* seedlings.

Comment #3: I find it difficult to understand the results shown in Fig. 7a. *GCHY5* itself can restore the long-hypocotyl phenotype of *hy5* to varying extents (Fig. 6c) and play dominant roles over *GCHXK* (Fig. 7a), it is hard to believe that *GCHY5*'s activity in *GCHXK/GCHY5/hy5* is strikingly restrained as presented in Fig. 7a. Did the authors examine the expression levels of *GCHXK* and *GCHY5* in different genotypes used in this figure?

Response: This result was surprising at first, but was replicated and confirmed in additional experiments. The results of this experiment indicate that having *HY5* only in the guard cells is insufficient to inhibit the elongation induced by *GCHXK* and that the inhibition of elongation by *GCHY5* requires the presence of *HY5* in tissues other than guard cells. It is also known that *HY5* can auto-activate its own expression (Ref's within the text), a situation that does occur in the case of *GCHXK/GCHY5/WT*, but not in the *GCHXK/GCHY5/hy5*. The expression analysis of *HXK1* and *HY5* in the different genotypes is now provided in Fig. 6B-C.

Comment #4: The authors claimed that *GCHY5-GFP* proteins can be exported to mesophyll cells in the presence of light (Fig. 8-10). Since *HY5* proteins are degraded in the dark and stabilized in the light, the steady state level of *HY5* proteins would be expected much lower in the dark than in the light. Therefore, alternative

interpretation of the results in Fig. 8-10 would be that a small portion of GCHY5-GFP proteins could move out of guard cells regardless of light, which are below the detection limit under microscope in the dark due to protein degradation. New experiment may be needed to clarify this point, for example, pre-treat the GCHY5-GFP seedlings with MG132 in the dark to block HY5 degradation, then examine the location of GCHY5-GFP proteins in the same condition.

Response: We thank the reviewer for this comment. Following the reviewer's suggestion, we conducted an experiment using the MG-132 proteasome inhibitor and analyzed the presence and intensity of HY5-GFP (Fig. 8). The presence of HY5-GFP outside the guard cells was observed in seedlings grown under light and in the dark. However, the HY5-GFP signal in the seedlings grown in the dark was half as strong as the one observed in the seedlings grown in the light. This result suggests that HY5-GFP exits the guard cells even in the dark, but is probably targeted for degradation by the proteasome. A detailed explanation of this experiment and its conclusions were added to the Results and Discussion sections.

General response: In several cases, figures were changed from bar plots to box plots, according to the editorial policy checklist.

References:

- 1 Stewart, J.L., Maloof, J.N. & Nemhauser, J.L. PIF genes mediate the effect of sucrose on seedling growth dynamics. *PLoS One* **6**, e19894 (2011).
- 2 Lilley, J.L.S., Gee, C.W., Sairanen, I., Ljung, K. & Nemhauser, J.L. An endogenous carbon-sensing pathway triggers increased auxin flux and hypocotyl elongation. *Plant Physiol.* **160**, 2261-2270 (2012).
- 3 Chen, X. *et al.* Shoot-to-root mobile transcription factor HY5 coordinates plant carbon and nitrogen acquisition. *Curr. Biol.* **26**, 640-646, doi:10.1016/j.cub.2015.12.066 (2016).

REVIEWERS' COMMENTS:

Reviewer #1 (Remarks to the Author):

In the Revised Manuscript by Kelly et al., the authors have tried to address my concerns with either new data or with suitable explanations. While many of them are satisfactory, i still have the following minor concerns/suggestions:

1. I sugest the authors to put WL, RL, BL data (hypocoty length and the pictures) together in the main figure 1. This will allow the readers for easy navigation of the text and results, and to undersatnd the consequences.
2. While PIFs have major function in Red light, the HXK effect is seen specifically in Blue-light, how can the autotros say that HXK response is due to increased PIFs? Is it that PIFs are getting induced in response to BL? but not to RL in a HXK dependent manner?

Rest is fine.

Reviewer #3 (Remarks to the Author):

This revised manuscript is much improved by adding new experimental results. Some of the figures are rearranged which are very clear now. All my concerns are fully addressed. Therefore I would recommend publishing the revised manuscript, as the well-presented data are of high interest for the community.

REVIEWERS' COMMENTS

Reviewer #1 (Remarks to the Author):

In the Revised Manuscript by Kelly et al., the authors have tried to address my concerns with either new data or with suitable explanations. While many of them are satisfactory, i still have the following minor concerns/suggestions:

Comment #1: *1. I sugest the authors to put WL, RL, BL data (hypocoty length and the pictures) together in the main figure 1. This will allow the readers for easy navigation of the text and results, and to undersatnd the consequences.*

Response: According to the reviewer's suggestion, we moved the results of Fig. S5 to the main text (Fig 2f and 2g). Fig. S5 was removed from the MS, and the text was changed accordingly.

Comment #2: *While PIFs have major function in Red light, the HXK effect is seen specifically in Blue-light, how can the auotros say that HXK response is due to increased PIFs? Is it that PIFs are getting induced in response to BL? but not to RL in a HXK dependent manner?*

Response: PIF's plays a significant role in the responses to both red and blue lights. The importance of PIF's for the response to blue light, in particular, is well established. Low blue light promotes the accumulation of PIF4 protein to support hypocotyl growth (Boccaccini et al., 2020). In addition, earlier work by Pedmale et al (2016) demonstrated that PIF's interact directly with CRY photoreceptors to mediate hypocotyl elongation in response to blue light (Pedmale et al., 2016). We, therefore, suggest that HXK promotes the induction of PIF's under blue light and that under red light, the level of PIF's is determined by other factors that do not involve HXK. The shared elongation response observed for GCHXK in response to white and blue lights occur since the white light spectrum comprises blue light wavelengths.

Rest is fine.

Reviewer #3 (Remarks to the Author):

This revised manuscript is much improved by adding new experimental results. Some of the figures are rearranged which are very clear now. All my concerns are fully addressed. Therefore I would recommend publishing the revised manuscript, as the well-presented data are of high interest for the community.

*General comment: We would like to thank the reviewers for their comments that have greatly improved this manuscript.

Refs.

Boccaccini, A., Legris, M., Krahmer, J., Allenbach-Petrolati, L., Goyal, A., Galvan-Ampudia, C., et al. (2020). Low Blue Light Enhances Phototropism by Releasing Cryptochrome1-Mediated Inhibition of PIF4 Expression. *Plant Physiol* 183(4), 1780-1793. doi: 10.1104/pp.20.00243.

Pedmale, U.V., Huang, S.C., Zander, M., Cole, B.J., Hetzel, J., Ljung, K., et al. (2016). Cryptochromes Interact Directly with PIFs to Control Plant Growth in Limiting Blue Light. *Cell* 164(1-2), 233-245. doi: 10.1016/j.cell.2015.12.018.